# Water regulates the residence time of Benzamidine in Trypsin

Narjes Ansari [1], Valerio Rizzi [1] & Michele Parrinello [1] ✉

The process of ligand-protein unbinding is crucial in biophysics. Water is an essential part of any biological system and yet, many aspects of its role remain elusive. Here, we simulate with state-of-the-art enhanced sampling techniques the binding of Benzamidine to Trypsin which is a much studied and paradigmatic ligand-protein system. We use machine learning methods to determine efficient collective coordinates for the complex non-local network of water. These coordinates are used to perform On-the-fly Probability Enhanced Sampling simulations, which we adapt to calculate also the ligand residence time. Our results, both static and dynamic, are in good agreement with experiments. We find that the presence of a water molecule located at the bottom of the binding pocket allows via a network of hydrogen bonds the ligand to be released into the solution. On a finer scale, even when unbinding is allowed, another water molecule further modulates the exit time.

The process of ligand-protein unbinding is crucial in biophysics and its full understanding would enhance not only our knowledge but also benefit the design of new drugs. In particular, two quantities are of great relevance, the ligand binding free energy and the inverse of the residence time $k_{off}$[1]. Being able to compute reliably these two quantities would be of great help. Here we describe a strategy that makes it possible to calculate accurately both quantities. We demonstrate this assertion with a state-of-the-art simulation of the Trypsin-Benzamidine system[2–13] and we find that the unbinding process can be more complex than what one could have anticipated. Although Trypsin-Benzamidine is one of the simplest cases of protein-ligand systems, high-resolution crystallographic experiments have recently demonstrated the presence of an extensive water structure in the binding cavity[14]. Our study reveals that water has not only a structural role but a dynamical one, since it regulates the unbinding process via a complex rearrangement of hydrogen bonds (HBs). In particular, the presence of a water molecule at a specific position in the binding cavity allows the unbinding process to take place. As the ligand begins to leave its binding pose, the number of water molecules in the binding cavity increases, leading to a finer regulation in which water determines two possible escape pathways with a $k_{off}$ differing by one order of magnitude.

Several technical advances have made this study possible. Ligand unbinding is a rare event that takes place on a timescale of milliseconds and thus its study requires the use of an enhanced sampling method. Here, we profit from the flexibility and efficiency of the On-the-fly Probability Enhanced Sampling (OPES) method[15], that is the latest evolution of Metadynamics[16,17]. Like umbrella sampling[18] and other enhanced sampling methods[19], OPES is based on the use of a set of collective variables (CVs) that are functions $s(R)$ of the atomic coordinates $R$ and describe the slow modes of the system. A good choice of $s(R)$ is essential to obtain converged results in an affordable time. To determine good CVs we use two machine-learning-based tools that we recently developed, Deep Linear Discriminant Analysis (Deep-LDA)[20] and Deep Time-lagged Independent Component Analysis (Deep-TICA)[21]. In building these CVs, we shall pay great attention to the role of water and make use of the experience gained in ref. 22.

Standard OPES is very efficient in calculating static properties such as binding free energies, but, in doing so, it alters the natural dynamics of the system so that a sensitive quantity like $k_{off}$ cannot be easily extracted. Nevertheless, it has been pointed out that if an enhanced sampling method can be engineered such that no bias is added to the transition region, the value of $k_{off}$ can still be computed[23–27]. Here we show that by an appropriate setting of the input parameters, OPES can be made to satisfy this condition. We call this approach OPES flooding (OPES$_f$)[28] since it is inspired by the flooding approach[26] and use it to calculate the ligand residence time.

[1]Italian Institute of Technology, Via E. Melen 83, 16152 Genova, Italy. ✉e-mail: michele.parrinello@iit.it

## Results

### Enhanced sampling technique for estimating free energies: OPES

To accelerate the occurrence of binding and unbinding events, we use OPES[15]. This method allows the system to overcome kinetic barriers by transforming the original CV probability distribution $P(s)$ into a smoother and thus simpler to sample one $P^{tg}(s)$. In the variant of OPES that we use here, we transform the $s$ probability distribution $P(s) = \frac{\int \delta(s-s(R))e^{-\beta V(R)}dR}{Z}$, where $\beta = 1/k_B T$ is the inverse temperature, $V(R)$ the interaction potential and $Z = \int e^{-\beta V(R)}dR$ the partition function, into the smoother well-tempered distribution $P^{tg}(s) \propto [P(s)]^{1/\gamma}$ [17], where the parameter $\gamma > 1$ regulates the broadening of the target distribution. In standard metadynamics this transformation is achieved by iteratively building a bias potential $V(s)$, while in OPES one instead reconstructs the probability distribution $P(s)$ on-the-fly. At iteration step $n$, the probability distribution $P_n(s)$ is estimated as

$$P_n(s) = \frac{\sum_k^n w_k G(s, s_k)}{\sum_k^n w_k} \qquad (1)$$

where $G(s, s_k)$ are multivariate Gaussian kernels evaluated at every step $k$, the weights $w_k = e^{\beta V_{k-1}(s_k)}$ are computed from the bias at step $k - 1$. In turn, the bias is

$$V_n(s) = \left(1 - \frac{1}{\gamma}\right)\frac{1}{\beta}\log\left(\frac{P_n(s)}{Z_n} + \epsilon\right) \qquad (2)$$

where $Z_n$ is a factor that measures the configuration space thus far explored and $\epsilon$ is a very important parameter that controls the maximum bias that can be deposited in the system.

### Machine learning-based CVs: Deep-LDA and Deep-TICA

As in our previous work[22], we use the machine learning-based Deep-LDA method[20] to design effective CVs to be used in conjunction with OPES. The method is based on the time-honored Linear Discriminant Analysis technique[29] employed in classification applications. Our aim is to build a CV that is able to distinguish between two sets of data. In our case, data come from unbiased simulations of the system in the bound (B) and unbound (U) states.

In standard LDA, one optimizes Fisher's ratio $\frac{\mathbf{w}^T \mathbf{S}_b \mathbf{w}}{\mathbf{w}^T \mathbf{S}_w \mathbf{w}}$ to obtain the linear combination $s(R) = \mathbf{w}^T \mathbf{d}(R)$ of descriptors $\mathbf{d}(R)$ that best separates the two states. Fisher's ratio is written in terms of the scatter matrix $\mathbf{S}_b = (\boldsymbol{\mu}_B - \boldsymbol{\mu}_U)(\boldsymbol{\mu}_B - \boldsymbol{\mu}_U)^T$ and the within matrix $\mathbf{S}_w = \mathbf{S}_B + \mathbf{S}_U$, where $\boldsymbol{\mu}_B, \boldsymbol{\mu}_U$ indicate the average descriptors values and $\mathbf{S}_B, \mathbf{S}_U$ the descriptors variance matrices in the two states. The vector $\mathbf{w}$ that maximizes this ratio is the direction that optimally discriminates the states and provides the best-separated projection of the data in the one-dimensional $s$ space[30].

In Deep-LDA, the set of $N_d$ descriptors $\mathbf{d}$ is fed into a neural network (NN) that is trained by applying the LDA criterion to the last hidden layer $\mathbf{h}$ of the network. In analogy with LDA, a projection of the $N_h$ components of the last hidden layer produces the Deep-LDA CV $s = \mathbf{w}^T \mathbf{h}$. This CV, being by construction a non-linear combination of the original input descriptors $\mathbf{d}$, is more expressive than a simple linear combination and has been successfully applied to solve a number of problems[22,31,32]. As $s$ tends to produce sharp distributions, we apply the cubic transformation $s_w = s + s^3$ [22,33] to make the CV more easily applicable to enhanced simulations.

While often successful in driving a system forth and back between different states, a Deep-LDA CV does not encode any information on the transition state. This information could have been obtained if one had access to a long dynamical trajectory in which many state-to-state transitions did occur. In this case, a way of building a good CV, would have been to use the Time-lagged

Independent Component Analysis[34,35], in which one looks for the most slowly decorrelating modes. These slow modes can be found using the variational principle[36]. If the modes are expressed as a linear combination of descriptors, the variational principle leads to a generalized eigenvalue equation. As in Deep-LDA, one can apply TICA not only to a linear combination of descriptors but also to the last hidden layer of a NN. This greatly improves the variational flexibility of the solution and thus its quality. In its original formulation, this approach was meant to be applied to unbiased trajectories. However, McCarty and Parrinello[37] using a linear approach and later Bonati et al.[21] using a non-linear one (Deep-TICA) have shown how to extract useful CVs from biased simulations. The resulting Deep-TICA eigenfunctions encode the slow modes of the biased simulation used for training. Here, we are going to apply Deep-TICA to the set of $N_d$ descriptors on a converged OPES trajectory where the Deep-LDA CV was biased.

### Enhanced sampling technique for estimating residence times: OPES flooding

As discussed in the introduction, to calculate rates from a biased simulations no bias must be deposited in the transition region. In such a case, the physical residence time $t$ is related to the physical simulation time $t_{MD}$ by[23,24]

$$t = \langle e^{\beta V(s)} \rangle_V \, t_{MD} \qquad (3)$$

where $V(s)$ is the instantaneous bias potential and the acceleration factor $\langle e^{\beta V(s)} \rangle_V$ is computed as an average along the simulation. To ensure that the condition under which Eq. (3) is satisfied, we introduce a variation of OPES that we call OPES$_f$[28] that is inspired by the variational flooding method[26]. The condition for OPES$_f$ to allow calculating physical rates are easily satisfied. The maximum amount of bias deposited can be controlled by the choice of the $\epsilon$ parameter in Eq. (2), making sure that it is lower than the free energy barrier. Furthermore via the parameter `EXCLUDED_REGION`, one can prevent OPES$_f$ from depositing bias in a preassigned region of configuration space. Since, from an initial free energy surface (FES) estimate one can roughly estimate both the height and the location of the transition, the setup of a subsequent OPES$_f$ rate calculation is relatively simple and straightforward[28].

### Designing water CVs

In our OPES simulations we shall use as CVs the distance $z$ from the binding site and a CV that encodes water behavior. Water is known to play a non-trivial role in ligand binding[38-46], therefore we want to use the power of machine learning techniques to capture and encode its behavior in a CV, generalizing the strategy that we have proposed for a smaller host-guest system in ref. 22. The choice of an effective set of descriptors $\mathbf{d}$ is critical as it must capture the solvation in different situations that are relevant to the binding-unbinding process, i.e., the ligand itself, the binding position, the binding pocket, and the binding path.

Some of the descriptors can be identified following physical intuition. For instance, to describe ligand's solvation we focus on its charged tail and use the coordination number of water around the Carbon atom of the amidine group ({G}) (see Fig. 2c). Regarding the binding position, we analogously evaluate the coordination number between water and the Carbon atom of the carboxylate group in residue Asp189 ({H}). These choices are similar in spirit to other solvation variables that have been devised in the past[22,47-51].

However, we follow a different approach to describe water behavior around the binding pocket and along the binding path. The S1 binding pocket of apo Trypsin form (see Fig. 1) is known from high resolution experiments[14,52] to be characterized by the presence of a set of deeply buried water molecules. These water molecules have a

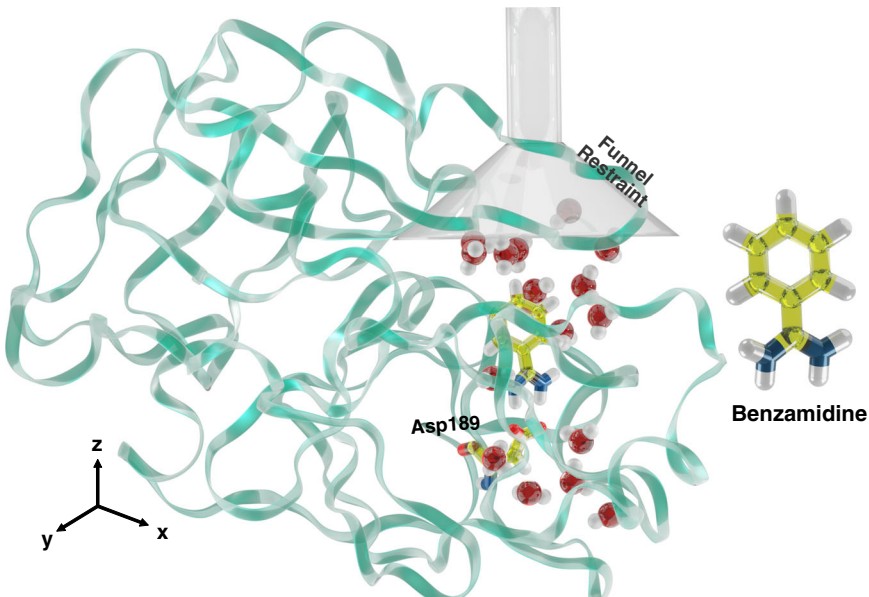

**Fig. 1 | The Trypsin-Benzamidine system.** A cartoon representation of Trypsin structure with the ligand Benzamidine and the Funnel restraint geometry. Oxygen, Nitrogen, Hydrogen, and Carbon atoms are colored in red, blue, white, and yellow, respectively. The protein is colored in green. A gray cone and cylinder represent the funnel restraint. The (un)binding path of the ligand is aligned along the $z$-axis. Relevant to binding residue Asp189 is highlighted.

long residence time and form what has been called in ref. 14 the reservoir. To encode their role in the descriptor set, we propose a general method that can be applied to describe trapped water molecules in biological systems. This method consists of four steps: (1) selecting the $\alpha$-Carbon atoms of the relevant part of the protein that encloses water (label 1 of Fig. 2a), (2) building the convex hull surface with the coordinates from step 1 (label 2 of Fig. 2a), (3) collecting the position of the water molecules that lie within the convex hull for longer than a pre-defined lifetime, and (4) clustering the collected data using a K-means clustering method to determine areas of high water density and their centers (label 4 of Fig. 2a). We call those points hydration spots ({$V_i$}) and use them as centers around which we calculate water coordination. The method is implemented in a Python script[53]. Following ref. 54, we also investigated the role of the ionic density and found that it does not play a relevant role in this system (see Supplementary Information (SI)).

In Trypsin, we restrict the analysis to the region around the S1 binding pose where we build the convex hull. That area encloses both the deeply buried water molecules and the binding path of the ligand (see Fig. 2b). From a number of unbiased trajectories (both in states B and U) we collect the positions of the water molecules that have a residence time inside the convex hull longer than 100 ps. Then, using the clustering method introduced in step 4, we identify 16 Vs (see Fig. 2c) positions. The number of clustering centers is chosen so that the relative distance between the Vs lies in a range of 2–3 Å. We find that there is a correspondence between the $V_5$–$V_{12}$ centers and the position of the reservoir water molecules reported in ref. 14. Furthermore, $V_3$ lies at the position of the long-lived water molecule that stabilizes binding and that in ref. 14 is called W1. We use as descriptors all the water coordination on ({G}, {H}, {$V_i$}) to generate Deep-LDA $s_w$ and Deep-TICA $s_t$ water CVs.

**Static properties: Binding free energy, enthalpy, and entropy**
To calculate the binding free energy, we perform an OPES simulation using 32 walkers, biasing $z$ and the Deep-LDA water CV $s_w$, for a total simulation time of 3.2 μs. More details about the simulation are provided in the SI. Convergence is achieved as several binding and unbinding events occur in a quasi-static regime of the bias (see Supplementary Fig. 2). For comparison, in the SI, we present an analogous

simulation in which we bias only $z$. In that case, the sampling is much worse quality and convergence is not reached (see Supplementary Figs. 3–5). In Fig. 3a, we show the FES projection on $z$, calculated as a block average with an error bar that is within the line width of the plot. We calculate the free energy difference between the B and U states using the funnel correction in Eq. (4) obtaining a value of 6.36 ± 0.07 kcal/mol, in an embarrassing and certainly fortuitous agreement with the experimental value of 6.36 kcal/mol[55,56].

The high level of accuracy that the combination of OPES and good quality CVs, makes it possible to estimate separately enthalpy $\Delta U$ and entropy $\Delta S$ of binding. This is achieved by converging binding free energy calculations in a range of temperatures and using the relationship $\Delta F = \Delta U - T\Delta S$. In Table 1 we report our estimate for these thermodynamic quantities and observe that they are also in good agreement with experiments. Further details about these simulations are provided in the SI.

In Fig. 3b, we show the two-dimensional FES of binding projected on $z$ and $s_w$, along with a number of representative snapshots of the different states and their typical water arrangement. State B is the global minimum of the FES and corresponds to the protein-ligand crystallographic structure (e.g., PDB 3atl[57]). In line with experiments[14], the W1 water molecule connects the ligand to Trypsin residues Tyr228, and Ser190, forming a total of 3 HBs. Furthermore, in the reservoir region below residue Asp189 we observe on average the presence of 5 water molecules, in agreement with the X-ray structure[14]. State B1 is a less stable binding pose where the ligand is in the same configuration as B, but the reservoir contains on average one more water molecule. In state I, the reservoir region contains another extra water molecule that tends to bridge the amidine group of the ligand and the binding site of Asp189, thus weakening binding. In the fully dissociated state U, the binding cavity returns to its apo form in which the 5 water of the reservoir are in the experimental structure of the holo form. The space previously occupied by the ligand is replaced by about 4–5 water molecules. The fact that in the apo form the reservoir structure of the water is preserved underlines once more the relevance of this structure (see Fig. 3f).

Note that, as the ligand progresses towards the U state, the number of water molecules in the binding site increases, underlining the role of water throughout the whole unbinding process. In spite of

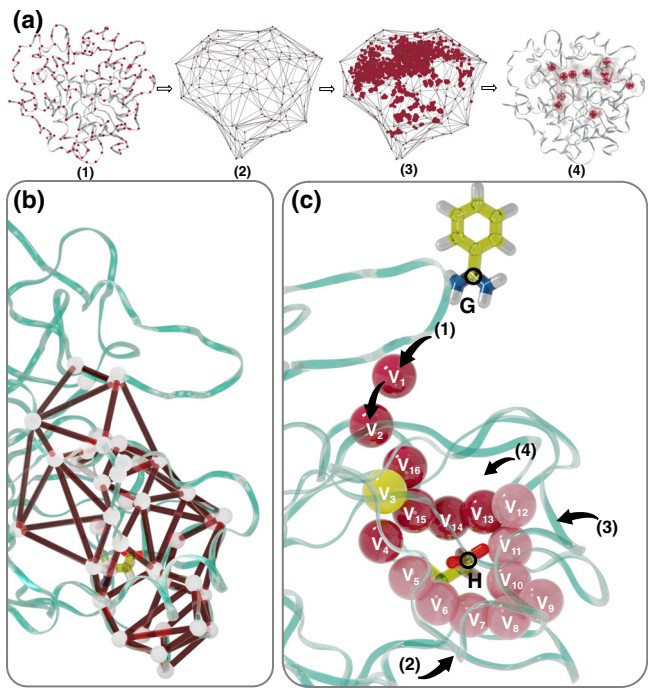

**Fig. 2 | Identification of long-lived water molecules. a** Graphical representation of the four steps strategy used to identify the long-lived hydration spots. (1) $\alpha$-Carbon atoms of the residues located on a protein's outer surface, represented by red dots. (2) Convex hull surface built from the $\alpha$-Carbon atoms shown in (1). (3) Distribution of the long-lived water molecules inside the surface. (4) Centers of the water distribution from step (3) obtained with a clustering algorithm. **b** Convex hull surface built around the binding pose of Trypsin. **c** Position of the 16 hydration spots $V_i$, shown by spheres. The hydration spots $V_5$–$V_{12}$ coincide with the position of the reservoir water molecules of ref. 14 and the dark red spheres lie on the binding path. The yellow sphere on $V_3$ shows the position of the key W1 water molecule[14] that forms a hydrogen bond with the ligand. Black arrows indicate the four possible paths of entrance/exit for water molecules in the binding pocket.

the limit of being trained on data coming exclusively from B and U, the Deep-LDA CV in combination with OPES is able to capture this non-trivial water behavior and converge the FES by virtue of the ability of OPES to deal with non-optimal CVs[58]. Nevertheless, $s_w$ is not able to resolve the diverse water arrangements in states such as B and B1, as we shall see below. In order to capture the finer details of the role of water in the binding pose, the intermediate states and the path to unbinding, we use Deep-TICA to determine a new water CV that we call $s_t$. Since Deep-TICA is trained on trajectories coming from biased simulations, the resulting CV is informed on the water modes that Deep-LDA is not able to resolve.

Using the data generated in the Deep-LDA-driven simulation, we project in Fig. 4a, b the FES along different pairs of CVs: $z$, $s_t$ and $s_w$, $s_t$, respectively. We find that $s_t$ resolves the original B state into states B and B', with state B being about 12 kJ/mol more stable than B' (see Supplementary Fig. 8). From these two states, two different unbinding pathways depart. We denote the states belonging to the less stable branch with a prime. Figure 4d, e shows typical configurations of the ligand and the surrounding water molecules in state B and B'. The number of water molecules in the reservoir is the same, but the water arrangement is different. In B, water molecules are part of an extended HB network that includes residue Asp189, while in state B' this network is less structured.

One striking difference between B and B' is that in state B there is one water molecule trapped deeply in the binding cavity, in an intermediate position between residues Tyr182, Lys219 and Gln219 which corresponds to descriptor V9 (see the semi transparent sphere in Fig. 4d). The FES projection along V9 and $s_t$ in Fig. 4c, confirms that V9 discriminates well between B and B'. Further analysis reveals that state B presents a slightly larger volume of the reservoir region than B' (see Supplementary Fig. 7), which, in turn, possibly facilitates the formation of an extended HB network. A study of the relative importance of the water descriptors on the NN CVs can be also found in the SI (see Supplementary Fig. 11).

The presence of a water molecule in V9 discriminates well between B and B', as can be seen, if we project the FES along V9 and $s_t$.

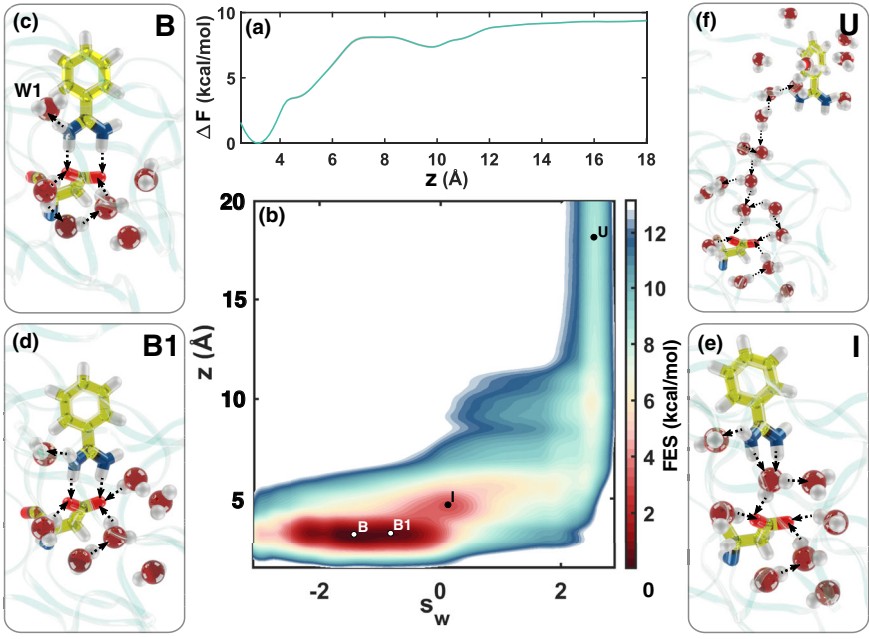

**Fig. 3 | Mechanistic interpretation of the binding process. a** 1D free energy surface of Trypsin-Benzamidine reconstructed using reweighting along $z$ with the statistical uncertainty within the line width. **b** 2D free energy landscape along $z$ and $s_w$ CVs. **c**–**f** Representative configurations of the relevant states are shown with a focus on the solvation pattern around the ligand and the binding pose.

**Table 1 | Our simulation estimates of the free energy, enthalpy, and entropy of binding at $T = 300$ K and corresponding experimental data[55,56]**

|  | $\Delta F$ | $\Delta U$ | $-T\Delta S$ |
|---|---|---|---|
| This work | $6.36 \pm 0.07$ | $4.12 \pm 1.56$ | $2.23 \pm 1.56$ |
| EXP. | 6.36 | 4.52 | 1.84 |

Thus this molecule stabilizes the cavity HB network that is such an important feature of this protein.

### Dynamic properties: Ligand residence time

We compute the ligand-unbinding rate by employing the OPES flooding technique[28] on simulations that start from state B. In this approach a choice of CVs that captures well the complexity of the path from state B to state U is essential. The Deep-LDA coordinate built only on the knowledge of the B and U states is not adequate to the purpose. In fact, being able to fill all the different metastable bound states is crucial for promoting transitions to the U state without remaining stuck in intermediate states. For this reason, besides $z$ we use Deep-TICA CV $s_t$. By construction, $s_t$ is able to extract the slow modes of the system and as a consequence to drive it out of deep basins towards the transition state. The use of OPES$_F$ for the calculation of rates requires that we define an excluded region to avoid depositing bias in the transition state region. An

analysis of the FES in Fig. 4a suggests to prevent depositing bias in the region $z > 6$ Å. We run a total of 55 ligand unbinding simulations and, by using Eq. (3), we determine for each simulation a physical ligand residence time $t$.

The distribution of transition times of a rare event dominated by a single barrier is expected to be Poissonian $\frac{1}{\tau}e^{-t/\tau}$ where $\tau$ is the characteristic time of the associated homogeneous process[59]. We fit all our data to such a model and find that the quality of the fit is poor (see Supplementary Fig. 13), as indicated by its low $p$-value. It is known that complex biological processes present multiple timescales and are not expected to necessarily follow such a simple model[1,60,61]. This led us to further analyze the unbinding trajectories and to identify the presence of two possible unbinding mechanisms: a faster and a slower one. The key difference between the two is related in the water network arrangement in the binding pocket (see Fig. 5). All unbinding events start with the water reservoir acquiring an extra molecule and the system reaching state B1 (see Fig. 5b). Then two possibilities occur. In the faster case, the presence of a water molecule in the vicinity of W1 weakens the bond between the ligand and W1 (see Fig. 5c), which leads to another water molecule to bind to residue Asp189 (see Fig. 5d) and finally brings about the ligand-unbinding event. In the slower mechanism, W1 is stable and, as the reservoir increases its water content (see Fig. 5e), eventually one water molecule bridges the ligand and Asp189 (see Fig. 5f), driving the system towards unbinding. As reported in Table 2, both mechanisms fit well the homogeneous model and

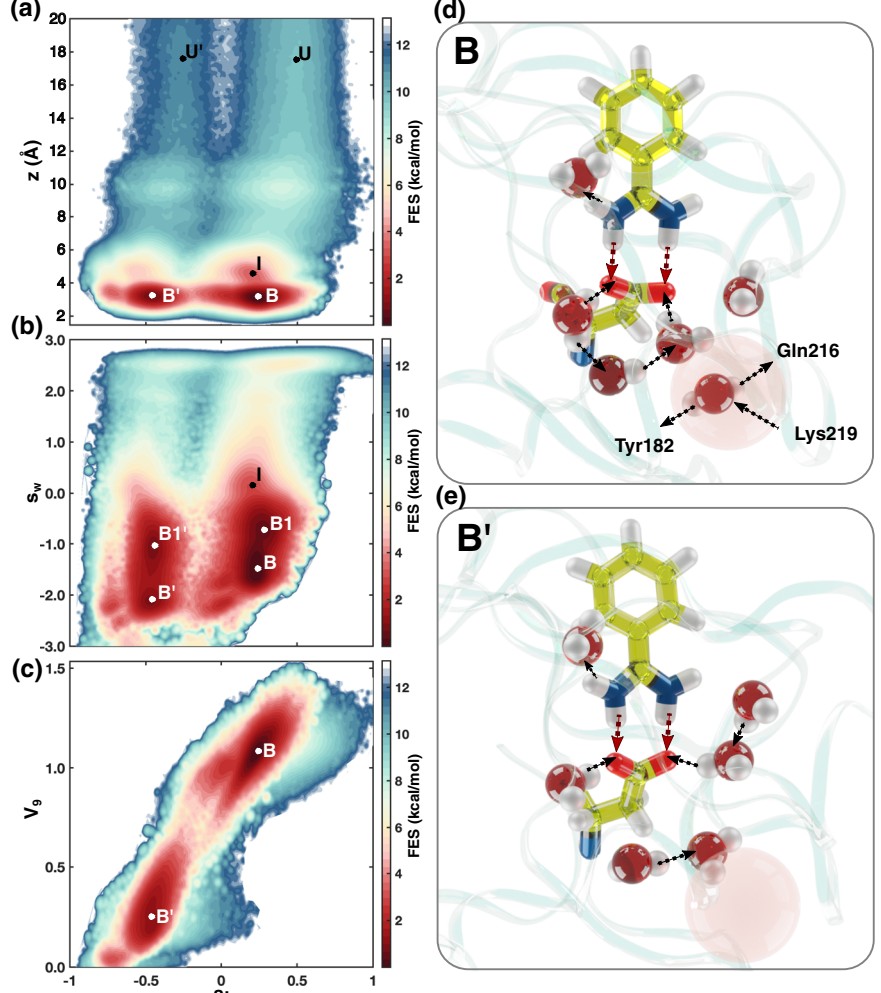

**Fig. 4 | Deep-TICA analyis.** 2D FES along $s_t$ and **a** $z$, **b** $s_w$, **c** V9. In **d** and **e**, representative configurations of the ligand and its solvation pattern in states B and B', respectively. The location of V9 is highlighted with a semi-transparent pink sphere.

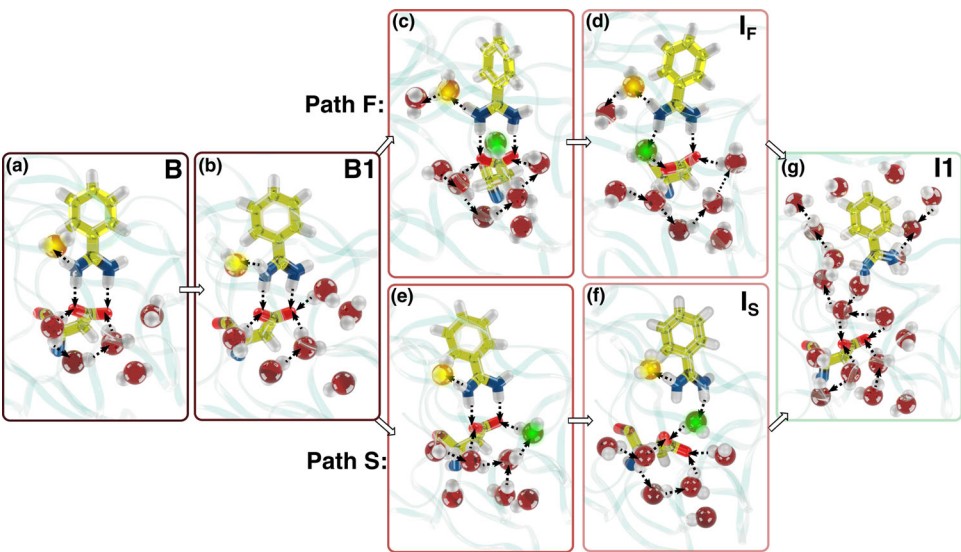

**Fig. 5 | Two ligand unbinding mechanisms.** Representative configurations of states **a** B and **b** B1. **c**, **e** The pre-intermediate step of fast and slow mechanisms, respectively. **d**, **f** The intermediate step of the faster ($I_F$) and the slower ($I_S$) unbinding mechanisms. **g** Intermediate $I_1$ that presents a number of water molecules between ligand and the binding pose. The yellow sphere highlights the location of the W1 water molecule. The green sphere indicates the position of the water molecule that bridges the ligand and the Asp189 residue. The protein is shown as a semi-transparent ribbon.

present a discrepancy in the resulting $\tau$ of about one order of magnitude. In particular, the slower mechanism $\tau$ that takes place in about 60% of the simulations has a $k_{off} = 1/\tau = 687$ s$^{-1}$ that is in close agreement with the experimental value of $k_{off} = 600 \pm 300$ s$^{-1}$ [62].

To further validate the pathways that we observe, we perform a set of OPES$_f$ simulations with an enlarged bias-free region $z > 4$ Å. These simulations ensure that the steps that lead from state B1 to state I occur in an unbiased regime (see Supplementary Fig. 14). We perform simulations where both $z$ and $s_t$ are biased and where only $z$ is biased. In both cases, we observe pathways belonging to the slow and the fast mechanism, in a similar proportion as the one from the full unbinding simulations. More details are in the SI.

## Discussion

Our work shows clearly that water plays a major role in the activation of Trypsin and modulates the release process of Benzamidine. Through the use of water-focused machine-learned CVs, we estimate static and dynamic properties in excellent agreement with experiments and we analyze the results with a level of resolution that lets us identify the importance of each individual water molecules. We observe how the presence of water in key positions affects the binding stability and determines unbinding mechanisms with significantly different ligand residence times. In conclusion, one might say that the set of reservoir water molecules, present both in the apo and the holo form, are an essential part of the enzyme, its structure, and its activity. We believe

that this is not just specific to the Trypsin-Benzamidine system and we argue that water would play such a non-trivial role at least in all the cases where the ligand has to negotiate its way out of a water-rich enzymatic cavity.

## Methods

We perform all simulations with the molecular dynamics code GROMACS 2020.4[63] patched with PLUMED 2.8[64] and the Pytorch library 1.4[20,65]. We use TIP3P water, while the protein is described by the Amber-14SB force field and the ligand interactions are taken from the Amber GAFF library[66]. We employ Ewald summation[67] for long-range electrostatic interactions with a cutoff of 10 Å for both the Coulomb and the van der Waals interactions. All simulations are run in the NPT ensemble with a timestep of 2 fs. We use the Parrinello-Rahman barostat[68] and the stochastic velocity rescaling thermostat[69], both with a coupling constant of 1 ps. For more details, see Supplementary Methods.

We train a Deep-LDA CV by running unbiased simulations on states B and U for 60 ns and evaluating the water coordination on a descriptor set **d** made by the 18 components ({G}, {H}, {$V_i$}). For training, we take as state B the initial configuration used in ref. 70. We use a NN made of 4 layers with $2.5 \times 10^{-5}$ learning rate. To train the Deep-TICA CV, we take the converged OPES trajectory used for calculating the binding free energy and feed the descriptors set **d** to a NN made of 4 layers with $1.0 \times 10^{-4}$ learning rate and 0.07 lag time. Further details can be found in the SI.

One of the difficulties in ligand-unbinding problem arises from the fact that once the ligand leaves the protein, it has to explore a large conformational space. To tackle this issue, we use the Funnel restraint proposed in ref. 71. In this method, the space available to the ligand in the unbound state is limited by confining it to a cylindrical volume above the binding site (see Fig. 1) which introduces an entropic restraint in the U state. To calculate the absolute free energy of binding, one needs to apply a correction to the apparent free energy $W(z)$ that comes from the simulation:

$$\Delta F = -\frac{1}{\beta} \log \left( C^0 \pi R_{cyl}^2 \int_B dz \, \exp(-\beta(W(z) - W_U)) \right) \quad (4)$$

### Table 2 | Ligand residence time

|  | $\tau$ (10$^{-3}$ s) | $k_{off}$ (10$^2$ s$^{-1}$) | p-value | $\mu$ (10$^{-3}$ s) | $\sigma$ (10$^{-3}$ s) | Number of events |
|---|---|---|---|---|---|---|
| All data | 0.64 | 15.6 | 0.06 | 1.10 | 1.49 | 55 |
| Faster path | 0.18 | 55.4 | 0.62 | 0.23 | 0.29 | 23 |
| Slower path | 1.45 | 6.87 | 0.63 | 1.58 | 1.59 | 32 |
| EXP. | – | 6.00 ± 3.00 | – | – | – | – |

$\tau$ is the characteristic time from an exponential fit[59], $k_{off} = 1/\tau$, and p-value measures the quality of the fit.

$\mu$ and $\sigma$ are the average and the standard deviation of the data, respectively. We show results from all our data and from data split into the two observed mechanisms. The experimental results are taken from ref. 62.

where $C^0 = 1/1660$ Å$^{-3}$ is the standard concentration, $z$ is the distance between the center of mass of the ligand and the binding site along the funnel's axis, $W(z)$ is the free energy along the funnel axis and $W_U$ is its reference value in state U.

## Data availability

All the inputs and instructions to reproduce the results presented in this manuscript can be found in the PLUMED-NEST repository at plumID:22.017. A tutorial on Deep-LDA training can be found at this link, while a tutorial on Deep-TICA is at this link. The python script used to determine areas of high water density and their centers can be found at this link.

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

## Acknowledgements

Part of this work was performed while the authors were associated with ETH Zürich and Università della Svizzera italiana, Lugano. The authors thank Faidon Brotzakis for providing the simulations initial input and they are grateful to Michele Invernizzi, Luigi Bonati, Jayashrita Debnath, Umberto Raucci, and Dhiman Ray for many useful conversations. The authors thank Michele Invernizzi for the implementation of OPES flooding. N.A. thanks Umberto Raucci for the support in developing the convex hull script. M.P. thanks Maria Ramos and Paolo Carloni for enlightening discussions on protein regulation.

## Author contributions

N.A. performed the simulations. N.A., V.R., and M.P. discussed the results and reviewed the manuscript.

## Competing interests

The authors declare no competing interests.
