## [Peer Review File · Nature Communications]

REVIEWER COMMENTS

Reviewer #1 (Remarks to the Author):

This manuscript presents a model of benzamidine unbinding from trypsin. It uses an enhanced sampling algorithm called OPES to capture this rare event. The authors show how both thermodynamic and kinetic information can be extracted from the model. Their results demonstrate an important role for water in determining the binding free energy and unbinding rate.

I enjoyed reading about all the methods the authors have devised and how they combined them in this study.

However, I do not think the work is a good match for Nature Communications. Much of the manuscript is a very technical review of the methods employed, most of which have been published already. The methodological advances are minor. The importance of water in general, and in this system in particular, is well appreciated. Value is added in this work, but it is not of sufficient novelty and broad appeal.

Reviewer #2 (Remarks to the Author):

This manuscript reports simulations of a protein-ligand binding process involving the well-studied trypsin/benzamidine system using a metadynamics enhanced sampling method. Based on these simulations, the authors conclude that certain water molecules in the binding pocket regulate the residence time of the ligand, which is the inverse of the rate constant for the dissociation of the ligand from the protein. The role of water molecules in the kinetics of protein-ligand dissociation has been an important and timely topic.

However, the above conclusion, which is also stated in the title of the manuscript, is fundamentally flawed and not supported by the results. While it is possible to indirectly estimate a kinetics observable (i.e. residence time) from metadynamics simulations by approximate “removal” of the bias that has been applied to the dynamics, the resulting protein-ligand dissociation pathways remain biased since the pathways have been generated on a modified free energy landscape. Thus,

it is not possible to draw any conclusions about the dynamics of these pathways that would be relevant to truly unbiased pathways. Furthermore, even if the pathways were generated on an unmodified free energy landscape, the results do not support the conclusion that the water molecules play an important role in the kinetics of protein-ligand dissociation – at a minimum, one would need to run simulations of the dissociation process with the water molecules removed.

In its current form, the primary strength of the manuscript is in the methodological details of the metadynamics simulations and techniques for characterizing the water molecules in the binding pocket. As such, the manuscript would mainly be of interest to the metadynamics community and would therefore be better suited for a more specialized journal.

Reviewer #3 (Remarks to the Author):

In this paper the authors present a workflow that combines two neural network analysis techniques (DeepLDA and DeepTICA) with the OPES enhanced sampling method to compute transition rates and free energies in the well-studied trypsin-benzamidine system. They examine a set of water-based features that describe hydrogen bond networks among long-lived water molecules. The authors show that these interactions are important for a complete description of the binding pathway. I enjoyed reading this paper and I think this is an interesting and important contribution to the field. However I have a number of critiques and questions, outlined below:

1) The Deep-LDA results are strange in that there isn't a lot of emphasis on the H and G descriptors. Intuitively, these could perfectly discriminate between the bound and unbound states. What were the accuracy values of the trained Deep-LDA models? Are these better than, say, a simple classifier based on H or G only?

2) Another puzzling result is why the Deep-TICA reaction coordinate was seemingly orthogonal to the (un)binding coordinate. Since TICA is supposed to pick out the longest timescale processes, one would think it would identify water-based features that are highly correlated with (un)binding. Was s_t the tIC with the longest timescale? Was there another tIC with a shorter timescale that correlated with (un)binding?

3) The authors should discuss in more detail what excluded regions were used in this case. A 2D plot of the final deposited bias would be helpful to confirm whether bias was added to the transition state.

4) Here, only the density of waters in any given position were used as features. Could the authors speculate as to whether including orientation information would improve or alter the results here?

5) Related, since the ligand is charged and forms a salt-bridge with the binding site, did any ions approach the vicinity of the binding site? Ion density has been shown to play an important role in transition paths (<https://doi.org/10.3389/fmolb.2022.858316>), which could be the case in this system as well.

6) At the end of the results section, the unbinding rates of two pathways were computed, one with an off rate of 687 s^{-1} and one with 5540 s^{-1} . It is stated that the slower mechanism will dominate the (observed) residence time. This does not seem correct. If there are two pathways, the path with the lowest barrier should be the most likely.

7) It is difficult to connect the analysis of the two pathways with the rest of the manuscript. From my understanding, even though the differences in paths F and S result from differences in water structure, they cannot be distinguished by s_t , and thus plotting them on Fig 4a or Fig 4b would result in two overlapping lines on the right side of the surface. Is this true? If so, are there other tICs that can distinguish between the paths?

8) Although the paper is convincing that water plays an important role in the binding process, it does not conclusively show that incorporating the water features allows for more efficient or more accurate calculations of rates or free energies. It would be nice to compare to “control” simulations that employed a similar amount of sampling, but only used the z coordinate. Or perhaps just include a discussion of whether this approach is expected to be more efficient or accurate.

Minor:

In Figure 1 the figure caption should describe the difference between the big waters and the little ones.

A schematic diagram or a flowchart would be helpful to summarize the approach here.

How does eq S-3 measure the level of correlation between the blocks? Couldn't blocks with the exact same weights still be uncorrelated?

In Figure 4A, the units of the z axis should be nm.

In equation 3, what does the subscript V denote?

Reviewer #4 (Remarks to the Author):

The work presented is an excellent study on dynamics ligand-protein binding and the role of water in the dynamics of the unbinding process. The choice of system is quite adequate given the breadth of work in this model. The discovery of two different mechanisms is quite impressive and crowns an excellent analysis into these complex dynamics.

The manuscript is written in an insightful way and I appreciate that the authors will make the scripts available. This is well needed to guarantee reproducibility (given the many steps and different algorithms used in these calculations). However, it would have been better to make these already available for the reviewing process. I strongly recommend submission with just two minor comments.

1- the clustering algorithm used for the water CVs has not been provided, or at least I couldn't find what was used.

2- the option "EXCLUDED_REGION" is included in the text with very little explanation. Given that OPES flooding is one of the most novel simulation aspects of this manuscript, one should add further detail possibly in the SI or in the text how it effectively operates. It is not completely clear if the selection of the excluded region is done prior to the calculation (meaning that the region is known beforehand) or if this is updated, if no bias at all is introduced or just an upper limit, is there a hard cut, etc...

Italian Institute of Technology

Prof. Dr. Michele Parrinello

Atomistic Simulations
Via E. Melen 83, 16152, Genova, Italy
+39 010 2897 439
michele.parrinello@iit.it
www.iit.it/web/atomistic-simulations

We thank all reviewers for their constructive comments and questions that help us to clarify our discussions and put our manuscript on a firmer ground. Below, we reproduce the reviewers comments in black and our response in blue for each question or comment.

Reviewer 1:

This manuscript presents a model of benzamidine unbinding from trypsin. It uses an enhanced sampling algorithm called OPES to capture this rare event. The authors show how both thermodynamic and kinetic information can be extracted from the model. Their results demonstrate an important role for water in determining the binding free energy and unbinding rate.

I enjoyed reading about all the methods the authors have devised and how they combined them in this study.

However, I do not think the work is a good match for Nature Communications. Much of the manuscript is a very technical review of the methods employed, most of which have been published already. The methodological advances are minor. The importance of water in general, and in this system in particular, is well appreciated. Value is added in this work, but it is not of sufficient novelty and broad appeal.

We disagree with the referee on this point. It is true that the issue of the role of water has been often raised. However, there is an element of strong novelty in our work. It is not the simple presence of this or that molecule but rather the way water is organized in the binding cavity that leads to different behaviors. To the best of our knowledge, this has not been pointed out earlier and may have important consequences on the way we think about many enzyme work.

Reviewer 2:

This manuscript reports simulations of a protein-ligand binding process involving the well-studied trypsin/benzamidine system using a metadynamics enhanced sampling method. Based on these simulations, the authors conclude that certain water molecules in the binding pocket regulate the residence time of the ligand, which is the inverse of the rate constant for the dissociation of the ligand from the protein. The role of water molecules in the kinetics of protein-ligand dissociation has been an important and timely topic.

However, the above conclusion, which is also stated in the title of the manuscript,

is fundamentally flawed and not supported by the results. While it is possible to indirectly estimate a kinetics observable (i.e. residence time) from metadynamics simulations by approximate “removal” of the bias that has been applied to the dynamics, the resulting protein-ligand dissociation pathways remain biased since the pathways have been generated on a modified free energy landscape. Thus, it is not possible to draw any conclusions about the dynamics of these pathways that would be relevant to truly unbiased pathways. Furthermore, even if the pathways were generated on an unmodified free energy landscape, the results do not support the conclusion that the water molecules play an important role in the kinetics of protein-ligand dissociation – at a minimum, one would need to run simulations of the dissociation process with the water molecules removed.

We are certainly aware that adding a bias does change the dynamics and this issue has been discussed in several publications by us and other groups. There is by now a vast literature that indicates that if the bias is properly engineered, the rate of transition between different metastable states can be computed, even in the presence of a bias. This point was made a long time ago in two almost contemporary papers by Voter (J. Chem. Phys. 1997, 106, 4665) and Grubmüller (Phys. Rev. E 1995, 52, 2893), the key being that no bias should be added to the transition state region. At a later stage, different strategies have been suggested to achieve this result (for example, Tiwary et al., Phys. Rev. Lett. 2013, 111, 230602; McCarty et al., Phys. Rev. Lett. 2015, 115, 070601; Debnath et al., J. Phys. Chem. Lett. 2020, 11, 5076) and several successful applications are reported. Here, for the first time, we use a variant of the OPES method to guarantee that indeed no bias is applied to the transition state region.

We observe that capturing long-lived water molecules in a CV greatly improves sampling. Therefore we find it reasonable to assume that water plays a key role in driving the system towards unbinding, hence our approach in including it as a CV in the rate simulations that lead us to accurately estimate the unbinding rate of the system. While the statement that the bias changes the dynamics is true, it is also true that a bias is most efficient when it gently pushes the system along the dynamical pathways. Thus, under appropriate conditions one could infer dynamical properties from bias run. However we believe that this letter is not the appropriate context in which to carry such a discussion. For this reason, we have repeated the study of the transition from B to I, putting ourselves rigorously in the conditions in which the flooding-type approaches work, that is no bias is added to the transition state region. Thus we perform another set of 20 simulations where the bias deposition is limited to the bound state (state B and B₁). We define the EXCLUDED REGION parameter above $z > 4 \text{ \AA}$ and the BARRIER parameter of 10 kJ/mol. In these simulations we either bias both z and s_t or z alone to separate the role of the water CV in this process.

In both cases, we observe pathways belonging to the slow and the fast mechanism, in a similar proportion as the one from the full unbinding simulations. In Fig. L-1 we show representative trajectories of these new simulations, highlighting the fact

that any point away from the bound state present no bias. We included these results in the main text and in the SI.

Figure L-1: Representative OPES_f trajectories with EXCLUDED REGION $z > 4$ Å where (a,b) z and s_t are biased and (c,d) only z is biased. The transparent yellow box shows the EXCLUDED REGION where the opes bias is equal to zero and the plots are colored with the bias. Trajectories (a,c) display the faster mechanism and (b,d) the slower one.

As to the suggestion of the referee to “run the simulations of the dissociation process with the water molecules removed”, we struggle to understand the remark. If the referee means that we start from a dry cavity, we would find this remark rather unphysical since water features prominently in the structure of the binding site of Trypsin-Benzamidine (Schiebel et al., Nat. Comm. 2018, 9, 3559). If instead, the referee meant that we run calculations without biasing any water CVs and only biasing the ligand-protein distance z , we did perform such a test for the transition from state B to I. In both case we observe the same two mechanisms

In its current form, the primary strength of the manuscript is in the methodological details of the metadynamics simulations and techniques for characterizing the water molecules in the binding pocket. As such, the manuscript would mainly be of interest to the metadynamics community and would therefore be better suited for a more specialized journal.

We disagree with the referee on this point, as in our work, there is a strong element of novelty that is of interest to a wide community. We do not propose a technique to characterize some water molecules but rather a study of the organization of water in the binding cavity that leads to the discovery different behaviors. To the best of our knowledge, this has not been pointed out earlier and may have important consequences on the way we think about many enzyme work.

Reviewer 3:

In this paper the authors present a workflow that combines two neural network analysis techniques (DeepLDA and DeepTICA) with the OPES enhanced sampling method to compute transition rates and free energies in the well-studied trypsin-benzamidine system. They examine a set of water-based features that describe hydrogen bond networks among long-lived water molecules. The authors show that these interactions are important for a complete description of the binding pathway. I enjoyed reading this paper and I think this is an interesting and important contribution to the field. However I have a number of critiques and questions, outlined below:

1. The Deep-LDA results are strange in that there isn't a lot of emphasis on the H and G descriptors. Intuitively, these could perfectly discriminate between the bound and unbound states. What were the accuracy values of the trained Deep-LDA models? Are these better than, say, a simple classifier based on H or G only?

H and G are indeed relevant descriptors in discriminating between states B and U. Nevertheless, using descriptors that discriminate states well is not a sufficient condition to produce an effective CV to bias the system. Knowledge of the system and its slow degrees of freedom is essential. If a CV classifies well but does not include information about important slow degrees of freedom of the system, it would be a good classifier but would not be able to push the system effectively between such states.

In our case, we have shown in the main text that water molecules in the reservoir region and long-lived water molecules such as W₁ have an essential role throughout the (un)binding process. Not including them in the Deep-LDA training would lead to a CV that cannot capture them and cannot accelerate their motion, therefore hindering sampling.

In Figure S-11 we show a ranking of the relevance of the descriptors based on the derivative of the Deep-LDA CV with respect to the descriptors (see Rizzi et al., Nat. Comm. 2021, 12, 93). This ranking provides a measure of the relevance of a descriptor in driving the system away from states B and U. In both cases, the sum of G and H descriptor weights is around 25 percent of the total weight over 18 descriptors, while the other 16 descriptors' weight amounts to 75 percent. We have added further details to the SI, stressing upon the importance of finding and including relevant long-lived water molecules in the CV training.

2. Another puzzling result is why the Deep-TICA reaction coordinate was seemingly orthogonal to the (un)binding coordinate. Since TICA is supposed to pick out the longest timescale processes, one would think it would identify water-based features that are highly correlated with (un)binding. Was s_t the tIC with the longest timescale? Was there another tIC with a shorter timescale that correlated with (un)binding?

TICA is indeed supposed to pick out the longest timescale processes. However, the Deep-TICA strategy that we use and that was recently introduced (Bonati, et al, PNAS, 118, 2021) picks out the longest timescale process of the *biased simulation* used in its training. Our Deep-TICA is trained on trajectories where we biased the Deep-LDA CV that was in turn trained in distinguishing the water presence between state B and U. Therefore, the resulting Deep-TICA CV focuses on the water modes that the Deep-LDA CV is not able to accelerate well. To further stress this important point, we have added a comment in the main text.

3. The authors should discuss in more detail what excluded regions were used in this case. A 2D plot of the final deposited bias would be helpful to confirm whether bias was added to the transition state.

We are currently preparing a manuscript on the OPES_f approach where we present the method and explain the choice of simulation parameters. The OPES_f strategy requires that one roughly determines the transition state region in CV space so that one can define an EXCLUDED REGION where the bias deposition is inhibited. The 2D FES in Figs. 3b and 4a of the main text suggest that the transition region lies $z > 6 \text{ \AA}$. In this regard, we set the EXCLUDED REGION as $z > 6 \text{ \AA}$.

Following the suggestion of the referee, we have created Fig. L-2. The left panel shows the trajectory of z in a typical OPES_f simulation. The right panel presents a 2D scatter plot of the same trajectory over the CVs z and s_t . Both plots are colored with the instantaneous value of the OPES bias, with black corresponding to the case where no bias is present. As required, the EXCLUDED REGION ($z > 6 \text{ \AA}$) is unbiased. We have included a figure and further clarifications in the SI.

Figure L-2: A representative trajectory of $OPES_f$ along (a) z and time, (b) z and s_t . The transparent yellow box shows the EXCLUDED REGION ($z > 6 \text{ \AA}$), where the opes bias is equal to zero. The color indicates the instantaneous bias.

4. Here, only the density of waters in any given position were used as features. Could the authors speculate as to whether including orientation information would improve or alter the results here?

The lifetime of buried water molecules ranges from ns to μs , while molecular rotations occur on a much faster timescale, typically ps. This observation indicates that the slowest water modes lie in the translation of water molecules towards and away from their long-lived positions rather than in their orientation. To corroborate this claim, we perform a water analysis inspired by the work of Gelenter, et al. (Communications Biology, 4, 338, 2021) where one evaluates the non-uniformity of orientation of water molecules by estimating the entropy parameter $\Delta\Gamma$

$$\Delta\Gamma = \sum_{i=0}^{N_{\text{bin}}} P(\cos(\theta)) \ln[P(\cos(\theta))] + \ln[N_{\text{bin}}] \quad (\text{L-1})$$

where θ is the angle between a water molecule's dipole moment and a fixed axis, $P(\cos(\theta))$ is a probability density and N_{bin} is the total number of bins in θ . Higher $\Delta\Gamma$ value correspond to more ordered water and a higher orientational preference.

We perform this analysis over water in the vicinity of each hydration spot $\{V_i\}$ in two sets of trajectories, unbiased and biased, where both state B and U are visited. We consider water molecules located within 2 \AA of $\{V_i\}$. θ is the angle between their dipole moment and the binding axis z and we discretize it with a spacing of 0.1 radians. In Fig. L-3, we show the value of $\Delta\Gamma$ over unbiased (state B and U) and biased simulations and observe an analogous trend between the two. This indicates that the biased simulations reproduce the water ordering from unbiased simulations, even if no explicit information about the water orientation is present in the biased CVs. We have added this analysis on the water molecules orientation in the SI Fig. S-12.

Figure L-3: Entropy parameter estimated over biased (red) and unbiased trajectories (yellow) on water molecules in the vicinity of the 16 $\{V_i\}$ hydration spots.

5. Related, since the ligand is charged and forms a salt-bridge with the binding site, did any ions approach the vicinity of the binding site? Ion density has been shown to play an important role in transition paths (<https://doi.org/10.3389/fmolb.2022.8>) which could be the case in this system as well.

In *Front. Mol. Biosci.* 9, 2022, Roussey and Dickson find that ionic density plays a relevant role in ligand (un)binding. Our simulations includes 7 Cl^- ions. To evaluate the ionic importance in our system, we measure their presence in the vicinity of the binding site and the ligand itself. We first count the number of Cl^- ions within 5 Å of the Carbon atoms of the carboxylate group in residue Asp189 in our 3.2 μs biased trajectory. We do not observe any presence of Cl^- , as the probability of finding a negative ion in the binding pose is severely hindered by the fact that residue Asp189 has a negative charge. On the other hand, the ligand Benzamidine presents a partial positive charge on its Carbon atoms. Analogously, we count the number of Cl^- ions within 5 Å of the Carbon atom of the Benzamidine amidine group and observe the presence of an ion within the cutoff only in $3.6 \times 10^{-5} \%$ of the trajectory. This observations lead us to believe that the ions do not play a relevant role in our ligand binding system. In other systems where the ions are more relevant, it would indeed be interesting and beneficial to capture them in a CV and accelerate them. We have added this analysis to the SI and added a comment in the main text.

6. At the end of the results section, the unbinding rates of two pathways were computed, one with an off rate of 687 s^{-1} and one with 5540 s^{-1} . It is stated that the slower mechanism will dominate the (observed) residence time. This

does not seem correct. If there are two pathways, the path with the lowest barrier should be the most likely.

We agree on this point. We removed the statement that the referee mentions and added further clarifications to the main text.

7. It is difficult to connect the analysis of the two pathways with the rest of the manuscript. From my understanding, even though the differences in paths F and S result from differences in water structure, they cannot be distinguished by s_t , and thus plotting them on Fig 4a or Fig 4b would result in two overlapping lines on the right side of the surface. Is this true? If so, are there other tICs that can distinguish between the paths?

The referee is right, the two pathways F and S cannot be distinguished by s_t and would indeed result in two overlapping distributions (see Fig. L-4a and L-4b for trajectories belonging to pathway F and S respectively). While biasing s_t helps in accelerating the system towards unbinding, we believe that the event that differentiates the two unbinding pathways (i.e. the movement of water in the vicinity of W_1) is largely unbiased. As the barrier underlying this event is not the dominant one in unbinding, it is not picked up by the first Deep-TICA eigenvalue which focuses mostly on water in the vicinity of V_9 . We verified that also the second Deep-TICA eigenvalue cannot distinguish this event (see Fig. Fig. L-4c and L-4d for trajectories belonging to pathway F and S respectively).

Figure L-4: Two representative trajectories of $OPES_f$ belonging to pathway F (panel a and c) and S (panels b and d). Panels a and b, show z and s_t . Panels c and d, show z and s_{t2} which is the second Deep-TICA eigenvalue. All plots are colored with the instantaneous value of the bias.

8. Although the paper is convincing that water plays an important role in the binding process, it does not conclusively show that incorporating the water features allows for more efficient or more accurate calculations of rates or free energies. It would be nice to compare to “control” simulations that employed a similar amount of sampling, but only used the z coordinate. Or perhaps just include a discussion of whether this approach is expected to be more efficient or accurate.

We find the suggestion of the referee very valuable. We have run a new simulation with a similar amount of sampling (around $3.2 \mu\text{s}$) biasing only the z CV with 8 walkers. We show the corresponding trajectories of this simulation in Fig. L-3. We have added the same figure to the SI in Fig. S-3.

Figure L-5: **OPES trajectories biasing only z** . Dynamics of z in the 8 walkers OPES simulation at 300 K where only z is biased. The plots are colored with the instantaneous value of s_w .

Due to the inefficiency of biasing only the z CV and not accelerating the water degrees of freedom, it is clear that sampling is severely slowed down as we do not observe any back and forth event. In Fig. L-4a we show the free energy of binding from this simulation using a block average (see SI for more details) or a single block at different simulation times. The lack of convergence is very clear. For comparison, in Fig. L-4b we show the analogous plots from the simulation in the main text where both z and s_w are biased.

In Fig. L-5, we show the binding free energy as a function of block number and time for both simulations. In the one where both z and s_w are biased, the value of the binding free energy strongly converges to a value of -6.3 kJ/mol. In the simulation where only z is biased, the free energy of binding is not converged. The data of the block average with the corresponding weights can be found in Table L-1. The weight distribution is very inhomogeneous as the

blocks do not contain a sufficient sampling of the phase space. Such weight distribution indicates the lack of convergence and the unsuitability to perform a block average in this case. Both figures and the table have been added to the SI in Figs. S-4 and S-5, and Tab. S-2.

Figure L-6: **Free energy surface comparison using different CVs.** FES calculated from simulations where we biased (a) only z and (b) both z and s_w . In both panels, the left hand side shows FESs from block analysis and the right hand side FESs evaluated at increasing simulation time.

Figure L-7: **Binding free energy comparison using different CVs.** Binding free energy ΔF (kcal mol^{-1}) calculated from simulations where we biased (a) only z and (b) both z and s_w . In both panels, the left hand side shows the value from block analysis and the right hand side FESs evaluated at increasing simulation time.

Table L-1: **Binding free energy block analysis using only z .** Binding free energy ΔF (kcal mol^{-1}) and its corresponding statistical weight w (a.u.) in every simulation block of the calculation biasing only z . It is clearly not converged.

Block	CVs z	
	ΔF	w
1	11.9	640
2	11.1	524
3	7.0	1.4
4	5.6	0.4
5	6.8	5.8
6	-0.3	0.0
7	1.2	0.0
8	3.7	0.2
9	0.8	0.0
all	11.5 ± 0.6	

Minor:

In Figure 1 the figure caption should describe the difference between the big waters and the little ones.

The difference in size in water molecules in that figure was only graphical. We changed the figure so that all the water molecules are the same size.

A schematic diagram or a flowchart would be helpful to summarize the approach here.

Following the suggestion of the referee, we summarize the method in the flowchart in Fig. L-6 that we also added to the SI as Fig. S-1..

Figure L-8: Flowchart of the method.

How does eq S-3 measure the level of correlation between the blocks? Couldn't blocks with the exact same weights still be uncorrelated?

The wording that we choose is unclear and we have rephrased it. Indeed, blocks with the same total weights can be uncorrelated. In fact, Eq. S-3 measures the quality of the weighted block average by looking at the weight distribution in the blocks. If the enhanced sampling simulation reaches a quasi static regime and explores equally well the phase space in every block, the corresponding block average quality is illustrated by $N_{\text{eff}} \rightarrow N_B$. On the other hand, if the blocks do not sample equally well the phase space or the simulation is not well converged, one obtains $N_{\text{eff}} \ll N_B$.

A good example would be the comparison presented in point 8 between two simulations of comparable length but different quality of sampling. In one both z and s_w are biased, in the other one only z is biased. Performing a block average with 9 blocks, the former gives a $N_{\text{eff}} = 8.4$ while the latter is much lower $N_{\text{eff}} = 2.0$. For more discussions on the effective sample size in OPES, we refer to Invernizzi et al., J. Phys. Chem. Lett. 2020, 11, 2731.

In Figure 4A, the units of the z axis should be nm.

The referee is right. We have corrected the axis units.

In equation 3, what does the subscript V denote?

V represents the bias potential. We have clarified this point in the main text.

Reviewer 4:

The work presented is an excellent study on dynamics ligand-protein binding and the role of water in the dynamics of the unbinding process. The choice of

system is quite adequate given the breadth of work in this model. The discovery of two different mechanisms is quite impressive and crowns an excellent analysis into these complex dynamics. The manuscript is written in an insightful way and I appreciate that the authors will make the scripts available. This is well needed to guarantee reproducibility (given the many steps and different algorithms used in these calculations). However, it would have been better to make these already available for the reviewing process. I strongly recommend submission with just two minor comments.

the clustering algorithm used for the water CVs has not been provided, or at least I couldn't find what was used.

We used a K-means clustering method. We have added this information in the main text and a link where one can download the corresponding script.

the option "EXCLUDED_REGION" is included in the text with very little explanation. Given that OPES flooding is one of the most novel simulation aspects of this manuscript, one should add further detail possibly in the SI or in the text how it effectively operates. It is not completely clear if the selection of the excluded region is done prior to the calculation (meaning that the region is known beforehand) or if this is updated, if no bias at all is introduced or just an upper limit, is there a hard cut, etc...

We thank the referee for pointing out the novelty of the OPES flooding used in this paper. In Fig. L-1 we show a representative trajectory of OPES flooding highlighting the fact that the bias is turned off in the transition state region. We are currently preparing a manuscript on the OPES flooding approach where we present the method and explain the choice of simulation parameters. The EXCLUDED REGION is indeed chosen prior to the rate calculation, given the knowledge of the free energy landscape from preliminary simulations. A rough estimate of the transition region position in the CVs space is sufficient. One can see from the 2D FES in Fig. 3b and 4a of the main text that the transition region is located at $z > 6 \text{ \AA}$. So we choose such limit for the EXCLUDED REGION. We have added further clarifications in the SI and included Fig. S-6.

REVIEWERS' COMMENTS

Reviewer #2 (Remarks to the Author):

I am pleased to see that the authors have strengthened the manuscript by including additional calculations to ensure no bias in the transition state region, providing a great example of “best practices” for such metadynamics studies.

However, I stand by my original comment that the manuscript is not a good match for Nature Communications. As written, the bulk of the manuscript remains a technical review of metadynamics techniques that have previously been published. Furthermore, insights in the manuscript about the organization of water are not of sufficient novelty for broad appeal as structural roles of water in the binding cavity of the well-studied trypsin/benzamidine system have already been reported by other simulation studies. As such, the manuscript would be better suited for a specialized journal.

Reviewer #3 (Remarks to the Author):

Thanks to the authors for doing such a thorough job responding to the reviewer questions. I think the comparison with the z-only simulations makes a very compelling case for this approach and for the importance of water-mediated interactions to unbinding pathways in general. Excellent work!

Reviewer #4 (Remarks to the Author):

I am satisfied with the answers to my remarks and the changes introduced.